# Immunohistochemistry for Claudin-4 and BAP1 in the Differential Diagnosis between Sarcomatoid Carcinoma and Sarcomatoid Mesothelioma

**DOI:** 10.3390/diagnostics13020249

**Published:** 2023-01-09

**Authors:** Lina Zuccatosta, Tommaso Bizzarro, Giulio Rossi, Graziana Gallo, Stefano Gasparini, Andrea Ambrosini-Spaltro

**Affiliations:** 1Pulmonary Diseases Unit, Azienda “Ospedali Riuniti”, 60126 Ancona, Italy; 2Operative Unit of Pathologic Anatomy, Azienda USL Della Romagna, Infermi Hospital, 47900 Rimini, Italy; 3Pathologic Anatomy Unit, Fondazione Poliambulanza, 25124 Brescia, Italy; 4Operative Unit of Pathologic Anatomy, Azienda USL della Romagna, “Bufalini” Hospital, 47521 Cesena, Italy; 5Department of Biomedical Sciences and Public Health, Polytechnic University of Marche Region, 60126 Ancona, Italy; 6Pathology Unit, Azienda USL della Romagna, Morgagni-Pierantoni Hospital, 47100 Forlì, Italy

**Keywords:** mesothelioma, carcinoma, pleura, sarcomatoid, BAP1, Claudin-4, immunohistochemistry

## Abstract

(1) Background. In the differential diagnosis between sarcomatoid carcinoma (SC) and sarcomatoid mesothelioma (SM), we aimed to investigate the role of Claudin-4 and BAP1, a panel recently used to distinguish conventional carcinoma from epithelioid mesothelioma. (2) Methods. We collected 41 surgical pleural biopsies of SM, 46 surgical resections of SC from different sites and 49 pleural biopsies of normal/hyperplastic mesothelium. All the cases were tested for Claudin-4 and BAP1 using immunohistochemistry. The statistical calculations of the sensitivity, specificity and positive and negative predictive values were performed. (3) Results: Claudin-4 was negative in 41/41 SMs, while it was positive in 18/36 (50.1%) SCs (eight diffusely, 10 focally) within their sarcomatous component. BAP1 was lost in 23/41 SMs, while it was regularly expressed in 46/46 SCs. All the cases of the normal/hyperplastic mesothelium were negative for Claudin-4 and retained the regular expression of BAP1. The Claudin-4 expression was useful for detecting SC (sensitivity, 39.1%; specificity, 100%) and the BAP1 loss was useful for diagnosing SM (sensitivity, 56.1%; specificity, 100%). (4) Conclusions. The staining for Claudin-4 and BAP1 exhibited a low/moderate sensitivity in diagnosing SC and SM (39.1% and 56.1%, respectively), but a very high specificity (100%). Claudin-4 was expressed only in SC and BAP1 loss was noted only in SM.

## 1. Introduction

The diagnosis of mesothelioma with an epithelioid component is generally suspected based on the morphology and the special stains serve as a confirmation tool for medicolegal purposes [1,2]. We recently highlighted the discriminating role of a “two-hits” immunohistochemical panel consisting of Claudin-4 and BAP1 [3]; combining two or more different markers may represent the best compromise when using the special stains to distinguish mesothelioma with epithelioid features from conventional metastatic carcinoma. Nevertheless, one of the most difficult tasks faced with neoplastic pleural lesions is the distinction between sarcomatoid mesothelioma (SM) and metastatic sarcomatoid carcinoma (SC) [1,2,4,5,6,7]. In this setting, the conventional primary antibodies used to detect epithelioid mesothelioma and carcinomas have significant limitations [1,2,4,5,6,7,8]. GATA3 is the most recent and promising immunomarker to discriminate SM (diffuse and strong staining) from SC (negative or focal/patchy staining), with a 66% sensitivity and a 94% specificity [9,10,11]. However, Terra et al. observed a weak/moderate and heterogeneous expression of GATA3 in both tumors, suggesting a restricted role for this marker [12].

The loss of the nuclear expression of the breast cancer type 1 susceptibility protein (BRCA1) and the associated protein-1 (BAP1), a deubiquitylase that acts as a tumor suppressor molecule in mesothelioma and other neoplasms [13,14,15,16,17], and the loss of the cytoplasmic expression of methylthioadenosine phosphorylase (MTAP), a surrogate marker of the CDKN2A (p16) alteration, were recently tested in SM and SC with various results [17,18,19].

We experienced an excellent value of the “two-hits” panel of Claudin-4 and BAP1 in the differential diagnosis between mesothelioma with epithelioid features and conventional metastatic carcinomas, demonstrating a 94% sensitivity, a 100% specificity and a 96% diagnostic accuracy [3]. However, limited data have been reported for this panel regarding the distinction between SM and SC. The aim of the current study was to determine the role of the Claudin-4 expression and the BAP1 loss in the differential diagnosis of SM and SC and to compare our results with those of the previous studies.

## 2. Materials and Methods

We retrieved cases of sarcomatoid mesothelioma (SM) and sarcomatoid carcinoma (SC) in a retrospective and prospective study between 2006 and 2021 from our archival files. After an initial search using the local laboratory information system Armonia (Dedalus S.p.A., Firenze, Italy), the collected case series included surgical pleural biopsies of SM (41 cases) and surgical resections of SC from different primaries (46 cases). The diagnoses were reviewed using a multiheaded microscope by expert thoracic pathologists integrating the morphology and the immunoprofiles of the tumors with clinical data and imaging studies. As a control group, 49 pleural biopsies of normal mesothelium and benign reactive mesothelial proliferations were included. The cases were classified using the last WHO classification for thoracic tumors [20]. As suggested, pleomorphic carcinoma was used as a synonym for sarcomatoid carcinoma. We subdivided the SCs into monophasic sarcomatoid if they contained only the sarcomatoid component (spindle cells and/or giant cells) or biphasic if they also contained an epithelial component (adenocarcinoma, squamous cell carcinoma or carcinoma).

An immunohistochemical analysis was performed using Claudin-4 (monoclonal antibody 3E2C1, Thermo Fisher Scientific, Rockford, IL, USA) and BAP1 (monoclonal antibody C4/sc-28383, Santa Cruz Technology Inc., Dallas, TX, USA) in an automated immunostainer (BechMark ULTRA, Roche Diagnostics, Basel, CH, Switzerland).

The other primary antibodies variably used in this case series to reach the original report in the diagnostic routine practice consisted of prediluted primary antibodies from Ventana Medical Systems/Roche Diagnostics. They are listed as follows (with clones in parentheses): calretinin (SP65), podoplanin (D2-40), WT-1 (6F-H2), CK5/6 (D516/34), TTF1 (clone 8G7G3/1), CEA (clone PF3H8), Ep-CAM (Ber-EP4), CDX2 (EPR2764Y), CD10 (SP67), CK20 (SP33), CK7 (SP52), EMA (E29), estrogen (SP1), GATA-3 (L50-823), p40 (BC28), PSA (polyclonal), synaptophysin (MRQ-40), thyroglobulin (2H11+6E1), PAX8 (MRQ50) and chromogranin (LH2K10).

To quote a case as positive for Claudin-4, we considered the immunoreactivity of at least a weak staining in ≥5% of the tumor cells with a membranous localization with or without a cytoplasmic expression. We subdivided the Claudin-4 expression into (1) focal if limited to 5–9% of the entire neoplastic area and (2) diffuse when ≥10%. In the SCs with an epithelial component (biphasic SCs), we documented the Claudin-4 expression in both the epithelial and sarcomatous areas. The BAP1 staining was recorded only when it was restricted to the nuclei and the BAP1 loss was documented when no tumor cells showed the nuclear expression (in the presence of a consistent internal positive control). The statistical calculations included the sensitivity (number of true positives/true positives + false negatives), specificity (number of true negatives/true negatives + false positives), positive predictive value (PPV), negative predictive value (NPV) and diagnostic accuracy (DA). The statistical calculations were performed using the SPSS version 25 (IBM Corp., Armonk, NY, USA). This study was performed in accordance with the institutional ethical board protocols.

## 3. Results

The case series consisted of 46 SCs (27 biphasic with both epithelial and sarcomatous components and 19 monophasic sarcomatous with spindle and/or giant cells), 41 SMs (28 purely sarcomatoid, 13 desmoplastic), 33 samples with benign mesothelial hyperplasia and 14 with normal mesothelium. The majority of the SCs were from the lung (18 cases, 39.1%), but 28 cases were obtained from eight different primaries, namely the kidney (6), bladder (5), skin (6), pancreas (4), endometrium (3), colon (2) and one each from the breast and cervix.

Among the SCs, the overall expression of Claudin-4 was 65.2% (30/46 cases). In the monophasic SCs, the Claudin-4 expression was observed in 3/19 (15.8%) cases (in all three focal). In the biphasic SCs, the Claudin-4 expression was detected in 15/27 (55.6%) cases within the sarcomatous component (seven focal, eight diffuse) and in all 27/27 (100%) cases within the carcinomatous component (all 27 diffuse). Notably, 11/27 (40.7%) biphasic SCs showed a Claudin-4 positivity, with the expression confined within the carcinomatous component and the negativity of the sarcomatoid component (Figure 1). All the SCs maintained a nuclear immunoreactivity for BAP1 in both the carcinomatous and sarcomatoid components. Table 1 provides detailed information on the SCs and their immunohistochemical profiles.

In contrast, the SMs were all negative for Claudin-4 and showed a BAP1 loss in 56.1% (23/41) of the cases, varying from 53.6% (15/28) of sarcomatoid mesotheliomas to 61.5% (8/13) of desmoplastic mesotheliomas (Figure 2).

As expected, all the cases with normal mesothelium (14 cases) and reactive mesothelial proliferation (35 cases) were immunoreactive for BAP1 (no loss) and negative for Claudin-4. The detailed information on the Claudin-4 expression and the BAP1 loss is shown in Table 2.

To detect SC, the Claudin-4 expression in only the sarcomatous component showed a 39.1% and a 100% sensitivity. However, the Claudin-4 expression in all the components (epithelial and sarcomatous) showed a 65.2% sensitivity and a 100% specificity. To detect SM, the BAP1 loss showed a 56.1% sensitivity and a 100% specificity. The detailed data on the sensitivity, specificity, PPV, NPD and DA are summarized in Table 3.

## 4. Discussion

This study showed that the Claudin-4 expression and the BAP1 loss may help to distinguish between SC and SM. Even if the sensitivity was low to moderate, the specificity was very high for both the Claudin-4 expression and the BAP1 loss.

The differential diagnosis between SC and SM represents a real challenge at the morphological level and requires several immunostains to achieve a definite conclusion [1,2,4,5,6,7]. The initial studies demonstrated that a diffuse expression of GATA3 was indicative of SM [9,10,11], while a negative GATA3 staining favored SC of the lung with a 100% sensitivity and an 85% specificity. More recently, Piao et al. [10] demonstrated a GATA3 expression in 70.6% of SMs with an 83.3% specificity, but only in 16.7% of lung SCs. Terra et al. [12] confirmed that a diffuse expression of GATA3 favored SM over pulmonary SC with a 66% sensitivity and a 94% specificity. However, the patchy and focal staining observed in 51% of SCs limited the value of GATA3, if not associated with the other markers. In the same study [12], MUC4 immunoreactivity was noted in 3% of SMs and 38% of SCs of the lung, with a low sensitivity (38%) but a high specificity (97%) for the diagnosis of SC.

Since the loss of BAP1 and MTAP immunostaining is indicative of a malignant transformation in mesothelial cells and is frequently found in mesothelioma with an epithelioid component, several studies investigated the diagnostic role of these markers, even in its sarcomatous counterpart [8,15,17].

According to a study by Cigognetti et al. [13], the BAP1 loss was observed in 15% of SMs. The following authors reported similar findings. Terra et al. [19] demonstrated thte BAP1 loss in 10% (6/62) of SMs, where it was retained in all 31 pulmonary SCs. Hwang et al. [14] showed the BAP1 loss in 15% (3/20) of SMs (15%), where the BAP1 nuclear staining was retained in all 13 SCs from various sites (lung, bladder, pancreas, breast and colon). De Rienzo et al. [15] also reported the BAP1 loss in 15% (10/67) of SMs, a percentage significantly different from that reported for epithelioid mesothelioma (39%) and biphasic mesothelioma (33%). Kinoshita et al. [18] showed the BAP1 loss in a slightly higher number of cases: 36.7% (11/30) of SMs.

Surprisingly, Zaleski et al. [16] demonstrated the BAP1 loss in 92% (49/53) of SMs, a definitely higher rate than reported by other studies. In our study, the BAP1 loss was observed in 56.1% of SMs (61.5% in the desmoplastic type). The discrepancy in the BAP1 loss observed in SM may have different explanations, mainly related to the preliminary morphological selection of the tumors and the type of the analyzed specimen (small biopsy vs. surgical resection). The primary antibody clone (clone C-4) and the scoring system (complete absence of nuclear staining with positive normal cells as positive internal control) were identical in all the reported studies [13,14,15,16,17,18,19].

Claudin-4, a transmembrane tight junction and cell adhesion protein expressed in the normal epithelium and the majority of carcinomas, is not present in normal and neoplastic mesothelial cells [21,22,23]. Recent studies demonstrated that Claudin-4 is by far the most helpful negative marker of mesothelioma in terms of sensitivity and specificity, and the best positive pan-carcinoma antibody [3,23,24,25,26]. However, the role of Claudin-4 in SC remains poorly investigated. Ordóñez et al. detected the Claudin-4 expression in 30.7% (4/13) of pulmonary SCs, but they did not report any expression in 10 SMs [27]. Naso et al. [26] demonstrated the Claudin-4 expression in 33% (7/21) of pulmonary SCs, while they documented a complete negativity in all 31 SMs. In the current series, similar findings were observed. The Claudin-4 expression was found in 39.1% (18/46) of SCs from different primaries within the sarcomatous components and was not found in any of the 41 SMs. The literature data on the Claudin-4 expression and the BAP1 loss in SC and SM are summarized in Table 4.

In contrast to the other studies, even in a very limited series, Facchetti et al. reported a very high percentage of SCs with the Claudin-4 expression (83.3%, 5/6) [23]. However, they did not specify in which component (epithelial or sarcomatoid) the Claudin-4 expression was evaluated. In the biphasic SCs of our series, Claudin-4 was expressed in 15/27 (55.6%) cases within the sarcomatous component but was observed in all the cases within the carcinomatous component, which increased the percentage of SCs with the Claudin-4 expression from 39.1% (18/46) to 65.2% (30/46). Similar findings were observed by Ordóñez et al. [27], who described the consistent expression of Claudin-4 within the epithelial components of SCs. Even in anaplastic carcinomas with sarcomatoid components arising in ovarian mucinous tumors, Chaudet et al. demonstrated that the Claudin-4 expression was retained in 39% of anaplastic carcinomas, but was expressed within the mucinous component of all the tumors [28]. Considering the Claudin-4 expression only in sarcomatous or in the epithelial components leads to different sensitivities and specificities, as reported in Table 3. However, for research purposes, we found that it was more accurate to describe it only in the sarcomatous component and we used that value for comparison with the other studies in Table 4. Moreover, the Claudin-4 expression in the epithelial component of a biphasic lesion should also raise the possibility of biphasic synovial sarcoma [23,27].

A limitation of this study was the difference in the type and origin of the materials examined, i.e., the biopsies from the pleura in sarcomatoid mesotheliomas and the surgical specimens of the different primaries in sarcomatoid carcinomas. However, most SCs (18/46, 39.1%) were from the lungs, an organ that is frequently the primary origin of metastatic pleural lesions. Furthermore, the abundant neoplastic material in the surgical specimens of SCs led to reliable immunohistochemical findings, which would have been difficult to produce in small pleural biopsies of metastatic sarcomatoid carcinomas.

Based on the excellent sensitivity and specificity of this simple “double-hit” (Claudin-4 and BAP1) panel for discriminating mesothelioma with epithelioid features from conventional carcinoma [3], we replicated the same study design in the setting of sarcomatoid neoplasms, namely SM versus SC.

Taken together, our results mirror those reported by the other studies, indicating a more limited role of BAP1 and Claudin-4 in the differential diagnosis of SM from SC than observed in their corresponding epithelioid forms. However, the high specificity of the BAP1 loss in SM and the Claudin-4 expression in SC may contribute to a subset of these cases, particularly when used in combination with the other markers (i.e., GATA3 and/or MTAP).

## 5. Conclusions

From a diagnostic viewpoint, the Claudin-4 expression appears to be a helpful tool to favor SC and exclude SM. On the other hand, the BAP1 loss, although observed in a minority of the cases, remains a very specific finding to identify SM. Overall, the complete specificity of Claudin-4 in SC and the BAP1 loss in SM should be used in this differential diagnosis.

The expression of Claudin-4 and BAP1 do not have any prognostic role in SC and SM.

## Figures and Tables

**Figure 1 diagnostics-13-00249-f001:**
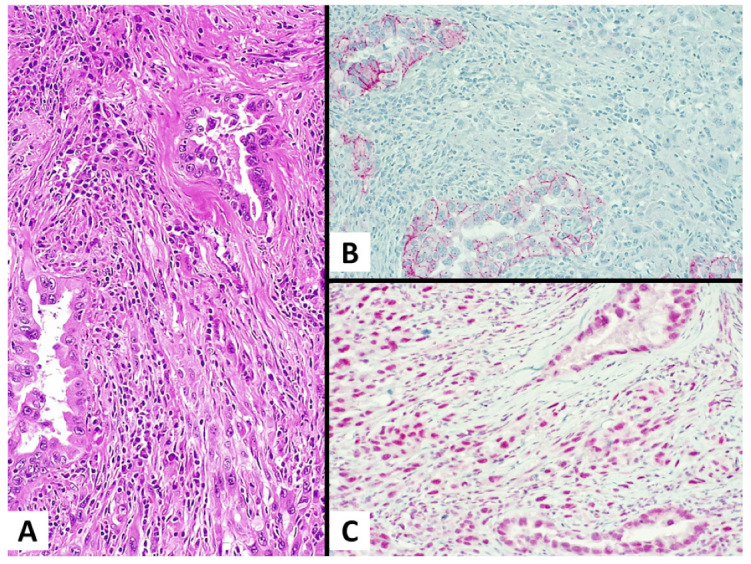
A case of sarcomatoid (pleomorphic) carcinoma of the lung. ((**A**) magnification ×200) Hematoxylin and eosin stain shows a sarcomatoid (pleomorphic) carcinoma of the lung consisting of an adenocarcinoma (epithelial component) and a spindle cell component (sarcomatous component); ((**B**) magnification ×200) The Claudin-4 expression is retained only in the epithelial component; ((**C**) magnification ×200) BAP1 is diffusely present (no BAP1 loss) in both components.

**Figure 2 diagnostics-13-00249-f002:**
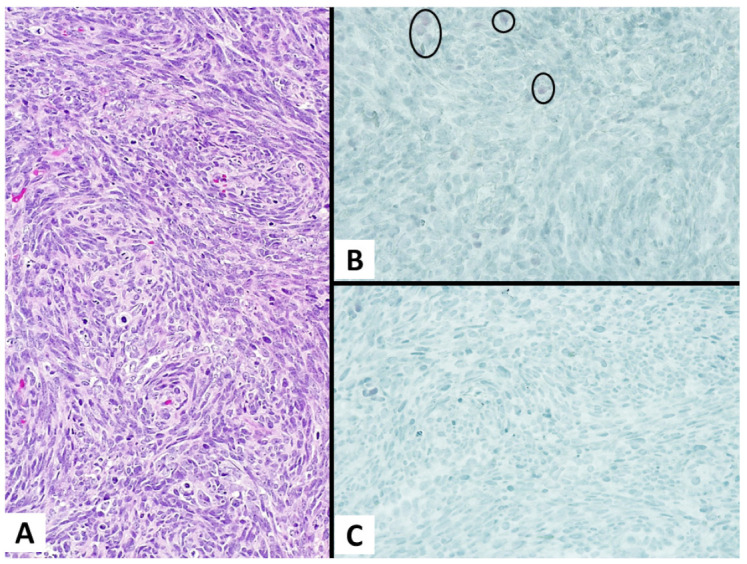
A case of sarcomatoid mesothelioma. ((**A**) magnification ×200) Hematoxylin and eosin stain shows a sarcomatoid mesothelioma consisting of a highly malignant spindle cell growth; ((**B**) magnification ×200) the tumor shows diffuse BAP1 loss (with some positive nuclei of lymphocytes into the circles as internal control) and ((**C**) magnification ×200) negativity for Claudin 4.

**Table 1 diagnostics-13-00249-t001:** Sarcomatoid carcinomas and distribution of Claudin-4 and BAP1 immunostaining.

Sarcomatoid Carcinomas		N. (%)	Claudin-4 Expression	BAP1 Loss
Total		46	18/46 (39.1%): -diffuse: 8/18 -focal: 10/18	0
Lung		18 (39.1%)	9/18 (50%)	0
	Sarcomatoid only	6	1/6	0
	Biphasic	12	8/12	0
Breast		1 (2.2%)	0	0
	Sarcomatoid only	1	0	0
	Biphasic	0	0	0
Renal		6 (13.0%)	3/6 (50.0%)	0
	Sarcomatoid	4	1/4	0
	Biphasic	2	2/2	0
Bladder		5 (10.9%)	1/5 (20.0%)	0
	Sarcomatoid only	2	0/2	0
	Biphasic	3	1/3	0
Pancreas		4 (8.7%)	2/4 (50.0%)	0
	Sarcomatoid only	1	0/1	0
	Biphasic	3	2/3	0
Colon		2 (4.3%)	1/2 (50.0%)	0
	Sarcomatoid only	0	0	0
	Biphasic	2	1/2	0
Skin		6 (13.0%)	2/6 (33.3%)	0
	Sarcomatoid only	5	1/5	0
	Biphasic	1	1/1	0
Endometrium		3 (6.5%)	0	0
	Sarcomatoid only	0	0	0
	Biphasic	3	0	0
Cervix		1 (2.2%)	0	0
	Sarcomatoid only	0	0	0
	Biphasic	1	0	0

**Table 2 diagnostics-13-00249-t002:** Distribution of Claudin-4 and BAP1 immunostaining in the entire series.

		N.	Claudin-4 Expression	BAP1 Loss
Sarcomatoid carcinoma	Total	46	30/46 (65.2%)	0
	Sarcomatoid only	19	3/19 (15.8%)	0
	Biphasic	27	-15/27 (55.6%) in sarcomatous component -27/27 (100%) in epithelial component -27/27 (100%) overall	0
Sarcomatoid mesothelioma	Total	41	0/41	23/41 (56.1%)
	Sarcomatoid	28	0/28	15/28 (53.6%)
	Desmoplastic	13	0/13	8/13 (61.5%)
Reactive mesothelial hyperplasia		35	35/35 (100%)	0
Normal mesothelial tissue		14	14/14 (100%)	0

**Table 3 diagnostics-13-00249-t003:** Statistics of the Claudin-4 expression and the BAP1 loss to detect sarcomatoid carcinoma and sarcomatoid mesothelioma, respectively. The expression of Claudin-4 was evaluated in the sarcomatous component only and in both components (sarcomatous and epithelial). *Legend*: PPV, positive predictive value; NPV, negative predictive value; DA, diagnostic accuracy.

Marker	Objective	Sensitivity	Specificity	PPV	NPV	DA
Claudin-4 expression in only sarcomatoid component	to detect sarcomatoid carcinoma	18/46 (39.1%)	41/41 (100%)	18/18 (100%)	41/69 (59.4%)	59/87 (67.9%)
Claudin-4 expression in all components	to detect sarcomatoid carcinoma	30/46 (65.2%)	41/41 (100%)	30/30 (100%)	41/57 (71.9%)	71/87 (81.6%)
BAP1 loss	to detect sarcomatoid mesothelioma	23/41 (56.1%)	46/46 (100%)	23/23 (100%)	46/64 (71.2%)	69/87 (79.3%)

**Table 4 diagnostics-13-00249-t004:** Literature review on the BAP1 loss and the Claudin-4 expression in sarcomatoid carcinomas. *Legend*: NA, not assessed; Ref. N.: reference number.

Author, [Ref. N.]	Sarcomatoid Carcinoma			Sarcomatoid Mesothelioma	
	Claudin-4 Expression * N (%)	BAP1 Loss N (%)		Claudin-4 Expression N (%)	BAP1 Loss N (%)
Cigognetti et al. [13]	NA	NA		NA	2/13 (15)
Hwang et al. [14]	NA	0/13	Primaries from lung, breast, colon, pancreas, bladder	NA	3/20 (15)
De Rienzo et al. [15]	NA	NA		NA	10/67 (15)
Terra et al. [19]	NA	0/31	Lung primary	NA	6/62 (10)
Ordonez et al. [27]	4/13 (30.7)	NA	Lung primary	0/10	n.a.
Naso et al. [26]	7/21 (33.3) **	NA	Lung primary	0/31	n.a.
Kinoshita et al. [18]	NA	NA		NA	11/30 (36.7)
Zaleski et al. [16]	NA	NA		NA	49/53 (92.5)
Facchetti et al. [23]	5/6 (83.3) **	2/13 (15.4)	Lung and breast primaries	0/11	NA
Present series	15/27 (55.6)	0/27	Primaries from lung, bladder, skin, breast, kidney, uterine cervix and endometrium, colon, pancreas	0/41	23/41 (56.1)

* The Claudin-4 expression is considered here within the sarcomatous component only. ** Not specified if the Claudin-4 expression is only located in the sarcomatous component.

## Data Availability

The data that support the findings of this study are available from the corresponding author upon reasonable request.

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
