# Peer review of "Immunohistochemistry for Claudin-4 and BAP1 in the Differential Diagnosis between Sarcomatoid Carcinoma and Sarcomatoid Mesothelioma"

_diagnostics, 2023, doi:10.3390/diagnostics13020249_

Round 1

Reviewer 1 Report

The paper is well done and conclusion shareable. Main limitation of the study is the surgical specimens of different primaries in sarcomatoid carcinomas. In any case,  this is well underlined in the discussion paragraph. I have only a few remarks:

-          In abstract are reported 36/36 SCs instead of 46

-          Table 2, data of Reactive mesothelial hyperplasia are not in line

Author Response

Authors’reply to the reviewer 1 comments.

The authors thank the reviewer for the favorable comments aimed at improving the manuscript.

  • Accordingly, we corrected the abstract inserting “46” instead of “36”.
  • Table 2 was modified according to the reviewer’s comment.

Reviewer 2 Report

Minor comments-

1.Provide scale bar for IHC images

2. Please rewrite the conclusion mentioning how the current study will impact diagnosis and prognosis in sarcomatous carcinoma and sarcomatous mesothelioma.

Author Response

Authors’reply to the reviewer 2 comments.

The authors thank the reviewer for the helpful comments.

  1. We provided the capture magnification of all the images inserted in the manuscript, including IHC stains.
  2. In agreement with the reviewer’s comment, the conclusion of the manuscript was re-written, better underlying the impact of the results of the study in terms of diagnosis and prognosis.